# SARS-CoV-2 Displays a Suboptimal Codon Usage Bias for Efficient Translation in Human Cells Diverted by Hijacking the tRNA Epitranscriptome

**DOI:** 10.3390/ijms252111614

**Published:** 2024-10-29

**Authors:** Patrick Eldin, Alexandre David, Christophe Hirtz, Jean-Luc Battini, Laurence Briant

**Affiliations:** 1Institut de Recherche en Infectiologie de Montpellier (IRIM), University of Montpellier, CNRS UMR 9004, 1919 route de Mende, 34293 Montpellier, France; 2Institut de Génomique Fonctionnelle (IGF), INSERM U1191, 141 Rue de la Cardonille, 34000 Montpellier, France; 3Institute for Regenerative Medicine and Biotherapy (IRMB)-Plateforme de Protéomique Clinique (PPC), Institut des Neurosciences de Montpellier (INM), University of Montpellier, CHU Montpellier, INSERM CNRS, 298 Rue du Truel, 34090 Montpellier, France

**Keywords:** SARS-CoV-2, codon usage, tRNA, epitranscriptome, translation

## Abstract

Codon bias analysis of SARS-CoV-2 reveals suboptimal adaptation for translation in human cells it infects. The detailed examination of the codons preferentially used by SARS-CoV-2 shows a strong preference for Lys^AAA^, Gln^CAA^, Glu^GAA^, and Arg^AGA^, which are infrequently used in human genes. In the absence of an adapted tRNA pool, efficient decoding of these codons requires a 5-methoxycarbonylmethyl-2-thiouridine (mcm^5^s^2^) modification at the U_34_ wobble position of the corresponding tRNAs (tLys^UUU^; tGln^UUG^; tGlu^UUC^; tArg^UCU^). The optimal translation of SARS-CoV-2 open reading frames (ORFs) may therefore require several adjustments to the host’s translation machinery, enabling the highly biased viral genome to achieve a more favorable “Ready-to-Translate” state in human cells. Experimental approaches based on LC-MS/MS quantification of tRNA modifications and on alteration of enzymatic tRNA modification pathways provide strong evidence to support the hypothesis that SARS-CoV-2 induces U_34_ tRNA modifications and relies on these modifications for its lifecycle. The conclusions emphasize the need for future studies on the evolution of SARS-CoV-2 codon bias and its ability to alter the host tRNA pool through the manipulation of RNA modifications.

## 1. Introduction: The Critical Role of Codon Bias in Translation Efficiency

The efficiency of genetic code translation into amino acid is of utmost importance for rapidly dividing organisms. This tedious process is carried out by the ribosome and necessitates accurate decoding of the genetic code in messenger RNA (mRNA), through the selection of transfer RNAs (tRNAs). The standard genetic code is degenerate: with the exception of tryptophan (Trp) and methionine (Met), most amino acids are encoded by two or more synonymous codons, which are used unequally within a genome (Figure 1A) [1]. This non-random distribution of synonymous codons, known as codon usage bias (CUB) [2], evolved alongside the expansion of the genetic code, shaped by evolution and natural selection [3,4]. CUB affects translation efficiency based on the availability of cognate codon-specific tRNAs, the effector molecules of translation, to ensure incorporation of the correct amino acids during polypeptide synthesis through complementary codon–anticodon base pairing [5,6]. CUB varies in a given genome. Highly expressed genes typically display stronger CUB, aligning with abundant tRNAs for optimal translation [7,8]. Conversely, rare codons may be selected on purpose, particularly at the beginning of coding sequences in eukaryotes. Indeed low CUB in the 5′ region decreases the translation elongation rate and reduces the likelihood of ribosomal traffic jams as translation progresses towards the 3′ end [9]. Additionally, reducing the elongation rate can promote the recruitment of chaperons facilitating co-translational protein folding [10,11]. Notably, CUB influences gene expression beyond translation, affecting transcription efficiency [12], termination [13], chromatin structure, and splicing [14]. As CUB varies depending on the organism [15,16], it is crucial for the investigation of the adaptation of infectious agents in their hosts, as well as virus evolution and pathogenesis. This report focuses on these aspects using SARS-CoV-2 as a case study.

In addition to the three codons that signal translation termination ((UAA, UAG, and UGA), 61 codons in mRNA are decoded through sequence complementarity with tRNA anticodons. Human cells display over 270 isodecoder genes (tRNAs with the same anticodon but different body sequence) among more than 610 annotated tRNA genes. Yet there are only 49 isoacceptor families (tRNAs with distinct anticodons but incorporating the same amino acid) to decode the 21 amino acids specified by the genetic code. Thus, the efficient deciphering of the 61 amino acid codons implies that some tRNAs can recognize more than one codon.

This conundrum led to the wobble hypothesis, introduced by Francis Crick in 1966 [17], proposing that only the first two bases of the codon pair precisely with corresponding bases in the anticodon, while the third position allows for flexibility or “wobble”. Accordingly, 30–40% of all codon recognition in a given organism is achieved through tRNA wobble recognition [18]. The modified wobble hypothesis of 1991 [19,20,21] expanded on the original hypothesis by including the role of certain base modifications occurring in or near the tRNA anticodon loop (Figure 1B). These modifications, ranging from simple methyl groups to more complex structures such as sugars, affect either the folding of the tRNA [22] or its ability to bind to codons [23]. Progressively, the impact of these modifications in almost all steps of RNA metabolism has emerged [24], and their potential consequences for translation fidelity are being characterized [21]. Currently, these modifications are recognized as major architects of the anticodon structure, capable of preventing, favoring, or expanding wobble base pairing, depending on their nature and position on the tRNA [25,26].

Recently, modifications in the tRNA anticodon loop, particularly at the U_34_ position of tRNA have emerged as a central pillar in controlling codon bias interpretation by the translational machinery, impacting translation fidelity [27]. Position 34 in the anticodon loop of tRNAs has indeed been identified as a hot spot for base modifications. The addition of a methoxycarbonylmethyl group at position 5 and a thiol group at position 2 (mcm^5^s^2^) of U_34_ in tRNA^Lys^_UUU_, tRNA^Gln^_UUG_, tRNA^Glu^_UUC_ and tRNA^Arg^_UCU_, by the stepwise action of the Elongator (ELP1-6), ALKBH8, CTU1/2 multi-enzyme complex [28] enhances base pairing with A-ending codons that are infrequent in the human genome (Lys AAA; Gln CAA; Glu GAA; Arg AGA codons) (Figure 1C). The first two enzymatic steps (Elongator and ALKBH8) generate mcm^5^ modifications at the U_34_ position of two additional tRNAs (tRNA^Gly^_UCC_ and tRNA^Arg^_UCU_) whereas Elongator alone can attach carbamoylmethyl (ncm^5^) to U_34_ of six different tRNAs (tRNA^Ala^_UGC_, tRNA^Thr^_UGU_, tRNA^Pro^_UGG_, tRNA^Ser^_UGA_, tRNA^Val^_UAC_, tRNA^Leu^_UmAA_). Additionally, KIAA1456 [29] (a human Trm9 homolog) has been recently shown to generate mcm^5^U directly from uridine [30]. In humans, additional modifications at position 34 include pseudouridine (Ψ), inosine (I), methycytidine (m^5^C), and queuosine (Q). Without these tRNA modifications, a codon-specific slowdown of translation occurs, impacting the overall translation efficiency of mRNAs enriched in corresponding codons [23,31,32].

### 1.1. Codon Bias and tRNA Pool

Translation efficiency cannot be explained solely by CUB, since preferentially used codons are not necessarily translated faster than non-preferred ones [33]. The overall control of translation efficiency also depends on the relationship between codon usage and the concentration of cognate tRNAs. In any given organism, the set of tRNA molecules, known as the tRNA pool, is fitted to the corresponding genome’s codon bias to ensure optimal translation. This mutual alignment, measurable by the tRNA adaptation index (tAI, defined below) [34,35], defines codon optimality, indicating how efficiently a codon is translated by the ribosome. Indeed, adapting CUB to the most abundant tRNAs decreases the time required for selecting the correct tRNAs, thereby reducing the likelihood of binding non-cognate tRNAs. In most prokaryotes and eukaryotes, there is a close correlation between tRNA level and the efficiencies of each codon–anticodon pairing. Consistent with translational selection, most optimal codons also have abundant corresponding tRNA copies in the human genome, although some still require wobble tRNAs. Yet, a close correlation between codon usage and tRNA abundance is not consistently observed, especially in multicellular organisms with higher tRNA gene redundancy [9]. In mammals, direct adaptation to anticodon pools does not fully explain synonymous codon usage, which is also shaped by mutational biases and genetic drift [34], such as GC-biased gene conversion [36]. Despite weak correlations between synonymous CUB and tRNA pools [37], the global correlation between codon and anticodon pools in mammals is strong and stable across different cell types and cell states [38,39]. Obviously, the extent of base modification in tRNAs also influences this balance especially for codons decoded through tRNA wobble recognition.

### 1.2. Viral Manipulation of Host Translational Machinery through tRNA Modifications

Viruses are obligatory parasites that rely entirely on the host’s translation machinery to translate their genome [40]. Numerous analyses comparing virus and host cell genomes in terms of codon usage have revealed that they use synonymous codons at different frequencies. Since a discrepancy between the viral codon usage and the availability of the corresponding tRNAs in the host tRNA pool induces ribosome pausing [27], the central question we would like to bring to light is how viruses overcome the inadequacy between their own CUB and the suboptimal tRNA pool composition of the cells they infect to efficiently translate their genome. In this context, viruses may need to maneuver host tRNAs to decode their skewed codons and optimize translation [41,42,43,44]. Moreover, because the tRNA pool in each cell type contains a subset of all the isodecoder and isoacceptor tRNAs needed for correct amino acid incorporation through complementary codon–anticodon base pairing, the differential expression of tRNA genes across tissues and individuals [45,46] may also influence viral tropism.

We here present a re-evaluation of SARS-CoV-2 coding sequences, specifically examining the congruence between viral codon preferences and the composition of the host’s tRNA pool. Our analysis, using a specialized codon analysis toolkit, found that SARS-CoV-2’s codon demand does not perfectly match the human host’s tRNA supply. Given that the codon bias was predominantly attributable to enrichment in A-ending codons, which necessitate RNA modifications of the cognate tRNAs at position U_34_ for efficient decoding, we hypothesized that the virus might manipulate the host’s tRNA epitranscriptome to circumvent this translational incompatibility. This hypothesis was subjected to experimental validation by investigating the impact of SARS-CoV-2 infection on tRNA modifications, revealing a substantial enhancement in U_34_ tRNA modifications upon infection. The additional observation that human cells unable to properly modify U_34_ tRNAs are no longer able to fully support the replication and the sustained spread of SARS-CoV-2, confirmed that SARS-CoV-2 infection efficiency depends on the accurate activity of the U_34_-tRNA modification pathway. The following schematic flowchart (Figure 1) outlines the process we used to generate our hypothesis and validate it experimentally.

**Scheme 1 ijms-25-11614-sch001:**
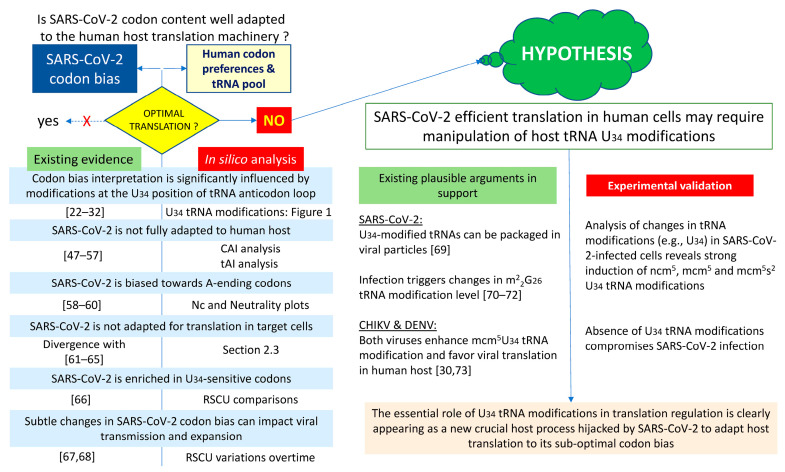
Visual representation of the sequential steps undertaken to develop and substantiate our hypothesis through experimentation. The left side details the supporting literature and original in silico analysis, which collectively led to the formulation of our central hypothesis. On the right side, this hypothesis was supported by existing arguments and further validated through our experimental approach [22,23,24,25,26,27,28,29,30,31,32,47,48,49,50,51,52,53,54,55,56,57,58,59,60,61,62,63,64,65,66,67,68,69,70,71,72,73].

## 2. Results and Discussion

### 2.1. Analysis of SARS-CoV-2 Codon Bias

Severe acute respiratory syndrome coronavirus 2 (SARS-CoV-2) is the causative agent of the recent devastating coronavirus disease 2019 (COVID-19) pandemic [74,75], which has infected over 600 million people and caused more than 6 million deaths worldwide (https://covid19.who.int, accessed on 4 October 2023). Coronaviruses belong to the order *Nidovirales*, with those infecting humans falling into two genera: alphacoronaviruses (HCoV-229E and HCoVNL63) and betacoronaviruses (HCoV-HKU1, HCoV-OC43, Middle East respiratory syndrome coronavirus (MERS-CoV), the latter of which includes the severe acute respiratory syndrome coronavirus (SARS-CoV1 and SARS-CoV-2). Since SARS-CoV-2 belongs to the same genus as SARS-CoV-1 and MERS-CoV, they share several structural characteristics [76] briefly outlined below [77,78]. 

SARS-CoV-2 is an enveloped virus with a positive-sense, single-stranded RNA genome of ~30 kb. Upon cell entry, two overlapping open reading frames (ORFs) ORF1a and ORF1b are translated from the positive strand genomic RNA (Figure 2A). The translation of ORF1b involves a −1 ribosomal frameshift enabling read-through of the ORF1a stop codon. ORF1a and ORF1b encode continuous polypeptides that are processed into 16 nonstructural proteins (Nsps) [79,80]. The viral genome is then used by the viral RNA-dependent RNA polymerase, Nsp12, to produce negative-strand RNA intermediates that serve as templates for the synthesis of positive-strand genomic RNA and subgenomic RNAs [81,82]. The subgenomic transcripts contain a common 5′ leader sequence fused to different segments from the 3′ end of the viral genome [83], along with a 5′ cap structure [84] and a 3′ poly(A) tail [85]. They encode four conserved structural proteins, namely, spike protein (S), envelope protein (E), membrane or matrix protein (M), and nucleocapsid protein (N), along with several accessory proteins. By homology with SARS-CoV1, SARS-CoV-2 is predicted to translate nine accessory proteins [86]. Nevertheless, the current annotation of SARS-CoV-2 (Reference Sequence: NC_045512.2) includes only six accessory proteins (3a, 6, 7a, 7b, 8, and 10), and not all of these ORFs have been experimentally reproducibly confirmed [87,88]. Using ribosome profiling techniques, the accurate quantification of canonical viral ORF expression was established, along with 23 novel unannotated viral ORFs [89]. These include several in-frame internal ORFs lying within existing ORFs, resulting in N-terminally truncated products, and internal out-of-frame ORFs producing novel polypeptides. 

#### 2.1.1. Codon Bias Analysis: A Tool to Shed Light on Virus History, Origins, and Evolution

It is well established that mutation pressure and natural selection are the primary factors shaping the codon usage of an organism [90]. Codon usage can also be influenced by nucleotide composition, synonymous substitution rate, gene length, expression level, and transfer RNA (tRNA) abundance [91,92]. Due to their reliance on host tRNA, viruses may evolve their codon usage to optimize or deoptimize translation in relation to their host’s codon usage [93,94]. Therefore, exploring the codon usage of viral genes is critical for uncovering viral evolutionary history [95], understanding virus–host interactions, and identifying the evolutionary forces shaping viral genomes [96,97]. Such information can also help characterize newly emerging viruses and trace their propagation across different host species. 

##### Codon Analysis Toolbox

The level of bias can be investigated according to the following widely used bioinformatic tools: 

*Codon Adaptation Index (CAI)*—The CAI estimates the degree of adaptation between a gene segment or an entire virus genome and a potential host [98,99,100]. It compares the codon usage in the viral sequence to a reference set of highly expressed genes from the host, which are assumed to use the most optimal codons for the host’s translational machinery. The CAI value ranges from zero to one, with one indicating that a gene uses the most frequently synonymous codon for each encoded amino acid and values close to zero indicating the use of less common synonymous codons. Higher CAI values between different genes on genome segments indicate a better adaptation to the host’s cell translational machinery. However, the CAI index is not strictly speaking a measure of CUB as codon usage is inherently multivariate and requires complementary approaches for comprehensive analysis. 

*Plotting the effective number of codons (Nc plot)*—Nc plots graph the effective number of codons used (Nc) against the G + C frequency at the third base position of the codon (GC3s). This quantifies how far a gene’s codon usage deviates from equal usage of synonymous codons [101]. Neutrality plots, which plot GC1 + 2 (mean G + C frequency at the 1st and 2nd position) against GC3 for each gene, are commonly used alongside Nc plots to estimate the respective contribution of mutation pressure and natural selection in shaping the CUB [102,103,104]. The slope of the curve in neutrality plots indicates the percentage contribution of mutual pressure to the overall codon bias.

*Relative Synonymous Codon Usage (RSCU)*—RSCU determines the intrinsic preference of a given cell or virus gene for synonymous codons by calculating the ratio of observed codon frequency to the expected frequency, assuming equal usage of all synonymous codons for the same amino acid [105,106]. The codon bias is considered positive for RSCU > 1.6 and negative for RSCU < 0.6, and unbiased for values in between.

*tRNA Adaptation Index (tAI)*—The tAI quantifies translational efficiency in a given context by considering the availability of tRNAs for each codon, factoring in wobble base-pairing efficiency (Wi) [107]. The normalized Wi values indicate the nominal speed at which the ribosome translates a codon relative to tRNA concentration, gene copies, and pairing efficiency. The tAI of a gene reflects the adaptation of its coding sequence to the intracellular tRNA pool, providing a measure of translational efficiency complementary to CAI, Nc, and RSCU [108]. If tRNA levels are not available, the tRNA gene copy number can be used instead [35,109] and retrieved from the dedicated database GtRNAdb [110] (http://gtrnadb.ucsc.edu, accessed on 22 June 2022).

#### 2.1.2. SARS-CoV-2 Adaptation to Various Species

Several reports based on sequence, phylogenetic, and recombination analyses suggest that SARS-CoV-2 originated from an ancestral coronavirus in bats, specifically related to the bat RaTG13 coronavirus [47], that likely passed through intermediate hosts such as pangolin *(Manis javanica)* before crossing species barriers again to infect humans [48,49,50,51]. We performed CAI [99] comparisons of the Wuhan reference strain NC_045512 of SARS-CoV-2 (weighted by the size of each ORF) against the codon usage table (CUT) of several species from the HIVE-CUTs database (https://hive.biochemistry.gwu.edu/cuts/, accessed on 13 September 2022). Our analysis confirmed a high degree of SARS-CoV-2 codon usage adaptation to both pangolin and bat (Figure 2B), supporting their proposed role as animal sources for this emerging virus. Interestingly, our analysis also revealed an even higher adaptation index to snakes (*Naja atra* among others) and to the marsupial wombat (*Vombatus ursinus*), suggesting these animals may also be relevant in understanding the virus’s transmission. Interestingly, the hypothesis that snake could serve as a potential intermediate host of SARS-CoV-2 between bats and humans has been debated [52,53,54] and remains plausible considering that snakes are a common wildlife meal in China and are ordinarily sold at the Wuhan seafood and animal market [55]. The wombat seems less likely since it lives only in Australia. However, the recent trend among wealthy Chinese people to hunt wombats in Australian wildlife reserves [56,57] makes it a tiny but potential virus transmission niche. Global CAI analysis of SARS-CoV-2 ORFs shows oscillations around the 0.69 mark (Figure 2A), indicating that none of the multiple ORFs are fully adapted to their human host codon preferences, with some regions showing poor adaptation with CAI values below 0.6, compared to CAIs of 0.869, 0.882, and 0.819 for the well-adapted highly expressed β-globin, β-myosin, and β-tubulin humans genes with protein abundance above 10,000 ppm, and CAI values below 0.78 for the poorly expressed RHA, RIG-I, and Kallmann Syndrome human genes (protein abundances below 300 ppm), respectively [111].

#### 2.1.3. Nc and Neutrality Plots 

If GC3s is the only determinant factor shaping codon usage, the Nc values would align with a dotted theoretical curve representing random codon usage [112] (Figure 3A,B). Assuming that (G + C) compositional constraints are the exclusive determinant of codon usage, GC3s and Nc values should exhibit a correlation that aligns with or deviates downward from the predicted expected trend. Throughout the spectrum of Nc values ranging from 20 to 61, a value of 20 signifies the highest degree of codon bias, while a value of 61 indicates no codon bias whatsoever. Substantial codon usage bias is typically indicated by an Nc value ≤ 35 in a coding sequence. The Nc–GC3s plot is widely used to distinguish whether codon usage in genes is primarily influenced by mutation (points near the expected curve) or also by other factors like selection (points significantly below the expected curve). When the Nc and GC3s values were plotted for the individual SARS-CoV-2 ORFs (Figure 3A, right), all points, except for ORFs 7a and 10, lied below the expected random curve, indicating that G + C compositional constraints might significantly influence SARS-CoV-2 codon usage. The deviation of almost all SARS-CoV-2 ORFs from the random curve towards the lower Nc values suggests that not only mutation but also other factors, such as translational selection, are likely to be involved in determining the selective constraints on codon bias in SARS-CoV-2 genes towards a preferred and limited set of codons. As a reference to the human genome, translational selection is much more pronounced in housekeeping genes [113] such as Globin, Myosin, and Tubulin than in poorly expressed human genes (DHX9, DDX58, IFN-beta, or KAL) as outlined in the corresponding Nc plot (Figure 3A, center). Accordingly, the weighted average for all SARS-CoV-2 ORFs (black dot with Nc = 45.1 and GC3s = 0.28 in Figure 3A) markedly differs from other RNA viruses, such as the *Flaviviridae* Zika virus ORFs (Figure 3A, left)*,* that also have a bias towards A-ending codons [58,59,60], showing a much lower GC content and overall Nc value for SARS-CoV-2, stressing again the unique codon-wise characteristics of SARS-CoV-2. The same analysis was performed for Orf1a and Spike gene segments of various coronaviruses (Figure 3B) including SARS-CoV-2 (Wuhan initial reference isolate and Omicron recent isolate), SARS-CoV-1, MERC-CoV, and various hCoV lineages (229E, OC43, and NL63). We observed that all points representing different strains were lower than the standard curve for both genes, Orf1a showing more dispersion on the Nc axis than Spike. Additionally, with the exception of SARS-CoV-2 Orf1a (Wuhan and Omicron), the coronavirus strains were not clustered together, highlighting again that mutational pressure combined with other factors may have contributed to the codon usage bias of SARS-CoV-2. The Spike Nc plot shows more clustering between strains, revealing less bias between strains for this gene segment, with a potential higher contribution of mutational pressure to the Spike codon bias. This was further confirmed with the complementary neutrality plot analyses (Figure 3C), which revealed the greater contribution of mutation pressure to Spike than to Orf1a CUBs (Spike:32% >> Orf1a:23%), while the relative constraints on GC3 (natural selection) being the main evolutionary force driving CUB is conversely higher for Orf1a (Orf1a:77% >> Spike:68%).

### 2.2. SARS-CoV-2 Genome Is Enriched in U_34_-Sensitive Codons

With the lack of information on the level of virus-mediated alteration of U_34_ tRNA modification in mind, we compared U_34_-sensitive codon frequencies in coronaviruses (HCOV-OC43, SARS-CoV-1, MERS-CoV, and SARS-CoV-2) and human genomes using the RSCU indicator [15]. A striking contrast emerged when these frequencies were arranged in clusters (Figure 4A). Codons preferred by SARS-CoV-2 (i) are barely used in human cells and (ii) predominantly include codons requiring U_34_ modifications on their cognate tRNAs for efficient decoding (with the exception of Gly (GGA)). Notably, SARS-CoV-2 exhibits a greater divergence from human codon frequencies compared to SARS-CoV-1 and MERS-CoV. Although the latter still shares some codon usage preferences with humans, SARS-CoV-2 contains a clear enrichment in U_34_-sensitive codons. This pattern was consistent across both nonstructural (Orf1a and Orf1b) and structural viral genes (Spike). U_34_-sensitive codon usage between SARS-CoV-2 and humans revealed a significant viral preference for U_34_-sensitive codons with up to two-fold enrichment for some codons such as Gln^CAA^, Arg^AGA^, and Leu^TTA^. (Figure 4B). 

#### 2.2.1. Comparison of Coronavirus Translation Adaptation (tAI)

Previous CAI measurements have the disadvantage of relying on highly expressed reference host genes. The translation adaptation index (tAI) offers a more nuanced approach as it can be based on either intracellular tRNA levels (when available) or tRNA gene copy numbers in the host genome. Here, we compared the tAI of various ORFs encoding non-structural (ORF1ab) and structural proteins (S, E, M, and N) among coronaviruses infecting humans. Our analysis revealed SARS-CoV-2 as the least adapted to the translational machinery of its human host, with all its ORFs having a tAI below 0.34, except for the Matrix protein (M) (Figure 5A). This tAI level obtained for SARS-CoV-2 is significantly lower than that observed for human ORFs encoding highly abundant proteins (such as β-myosin, β-globin, or β-tubulin). Calculations were performed using the human tRNA gene copy number retrieved from the genomic tRNA database (http://gtrnadb.ucsc.edu, accessed on 22 June 2022) known to mirror the global tRNA abundancy of a given organism but lacking information related to tissue specificity. However, it seems from the recent study by the Nedialkova group [114] that tRNA transcript levels may fluctuate without affecting significantly the tRNA anticodon pool abundance and availability that primarily dictate the decoding pace. Accordingly, the tRNA anticodon pool was shown to remain stable across cell types (human primary cells (cardiomyocytes (CMs), neuronal precursor cells (NPCs), and mature neurons) differentiated from iPSC cells) ensuring consistent decoding rates throughout development, independent of cell identity. This stability across cell types of tRNA pool was further authenticated by plotting normalized tRNA gene copy number (GCN) against the normalized experimental data of tRNA anticodon expression levels in these human primary cells (Figure 5B), showing a quasi-constant correlation between GCN and experimental tRNA levels in the four cell types. This observation validates the use of human tRNA GCN in the calculation of the aforementioned tAI used in Figure 5A.

#### 2.2.2. The Enigma of Spike Protein’s Furin Cleavage Site

Among the SARS-CoV-2 ORFs, the region encoding the spike protein has been extensively studied as it mediates attachment to the host cell by binding to the ACE2 membrane protein and facilitates viral fusion to the host cell membrane following efficient cleavage by furin proteases [115,116]. Early examination of the Wuhan SARS-CoV-2 isolate revealed an unusual furin-like cleavage site at the S1/S2 junction of the spike ORF [117,118]. This site, created by the insertion of a 4-amino acid sequence (PRRA), corresponds to the insertion of 12 nucleotides (...U CCU CGG CGG GC...) absent from all other sarbecoviruses, including the closely related bat sarbecovirus RaTG13 with ~96% genome sequence identity [74,119] (Figure 6A). In SARS-CoV-2, the furin site insertion lies in a region where codons are moderately adapted to the human host, as depicted by the CAI curve. Most codons in this region are commonly used in the virus genome (high RSCU), except for the arginine dicodon (CGG CGG (R R)). Interestingly, the corresponding unique CGG codon is less preferred than AGA in SARS-CoV-2 (SARS-CoV-2 RSCU_CGG_ = 0.17, SARS-CoV-2 RSCU_AGA_ = 2.63)), which is not the case in the human genome (human RSCU_AGA_ = 1.29, human RSCU_CGG_ = 1.21) (Figure 6C). During the pandemic, mutations impacting the furin site have been rare, suggesting a strong purifying selection in humans [120,121]. However, non-arginine residues in the PRRAR motif remain permissive to potential optimization during human viral evolution in different variants of concern and interest (Figure 6B). This complex interplay between codon usage and furin site evolution warrants further investigations to unravel its significance in SARS-CoV2’s origins. 

### 2.3. Suitability of the SARS-CoV-2 Highly Biased Codon Composition for Viral Translation in Target Tissues

The translation efficiency of the viral genome is heavily influenced by codon optimality, which is determined by the balance between the viral codon usage bias and the availability of a suitable tRNA pool in target cells. However, recent reports highlighted the significant variability in tRNA gene expression across human tissues [45,122,123,124]. This variability suggests the necessity of re-evaluating codon optimality by considering tissue-specific codon usage in compliance with virus tropism [61]. Whereas the evaluation of the effect of tissue-specific codon optimality on viral protein synthesis may remain experimentally elusive, a recent in silico study by Hernandez-Alias et al. [62] analyzing the relative codon usage landscape over 500 human-infecting viruses alongside tissue-specific tRNA expression profiles from 23 human tissues from The Cancer Genome Atlas (TCGA) has suggested that tRNA repertoires could determine tissue-specific translational efficiency [63]. They proposed that the SARS-CoV-2 proteome was well-adapted to tissues like the upper respiratory airways, lung alveoli [64], and gastrointestinal tract [65], seemingly matching the observed SARS-CoV-2 tropism. However, the translation appropriateness of these tissues was also matched by other viruses such as Flavivirus or Alphavirus that exhibit tropisms that do not share one shred of similarity with SARS-CoV-2 tropism. Several flaws in this study, including the use of tRNA abundance data derived from cancer cells in which profound deregulation of tRNA expression occurs [125,126,127,128], restrict its significance. Interestingly, recent studies showed that highly expressed genes in human lung primary tissue have a codon composition perfectly aligned with SARS-CoV-2’s codon usage, suggesting that the virus might have optimized its codon bias to take advantage of lung cells [129]. The recent Nedialkova group report [114] that illustrated the broad stability of tRNA anticodon pools across human cell types (shown above in Figure 5B), reinforces even more the prohibition of cancer cells in approaches aimed at tRNA pool dynamics. 

Taken together, these observations underscore the crucial need to further investigate how the codon composition of the viral ORFeome influences the translation rate of host genes and promotes viral translation.

#### 2.3.1. tRNA Modifications: Do RNA Viruses Have a Wobble? 

In analyzing virus–host interactions, it is crucial to consider the modification of host cell tRNAs promoted by the virus. While the functional importance of the virus mRNA epitranscriptome has been extensively reported in human cells [130,131] and more recently in the SARS-CoV-2 RNA genome and subgenomic transcripts [132,133,134,135], our understanding of virus-induced modifications in host tRNAs is still limited, although increasingly relevant [73]. As outlined earlier, modifications of the anticodon loop of tRNA regulate translation rate and fidelity, contributing to translational adaptation [136]. For instance, changes in tRNA modifications significantly impact translation efficiency in response to physiological stresses [137,138]. Hypomodification, occurring especially at position U_34_ of specific tRNAs, namely tRNA^Lys^_UUU_, tRNA^Gln^_UUG_, and tRNA^Glu^_UUC_, slows down translation, disrupts protein homeostasis, and reduces cellular fitness. These mechanisms are crucial for maintaining cellular function and viability during stress until normal conditions are restored. While these mechanisms may have a critical influence on viral translation, this aspect of virus/host relationships remains largely unexplored. 

#### 2.3.2. SARS-CoV-2 Codon Bias Dynamics during the Pandemic

In addition to providing crucial information on viral genetic adaptation to the host [66,139], evaluating nucleotide composition and CUB in the viral genome also provides insights into the mutational patterns of viruses over time and can be crucial for vaccines and antiviral therapy development [67]. A recent study analyzing over 3.5 million SARS-CoV-2 sequences revealed unique mutational trends with consistent nucleotide and codon frequencies [68]. This study also highlighted variations over time, including synonymous mutations, silent at the amino acid level, and nonsynonymous mutations inducing amino acid changes, impacting protein levels. It also revealed an unexpectedly high proportion of nonsynonymous mutations in the Spike gene when compared with glycoprotein genes from other RNA viruses. 

To deepen this analysis, we examined the sequences from the various SARS-CoV2 clades (Alpha, Beta/Mu, Delta, Lambda/Gamma, and Omicron) extracted from the study by the Fumagalli group [68] and generated their respective RSCU profiles monthly from December 2019 to July 2022. The RSCU patterns give us a snapshot of the average CUB for each month (Figure 7A). Focusing on SARS-CoV-2 Orf1ab and Spike genes, we analyzed the overall codon bias patterns by cluster analysis in order to compare the temporal codon usage variations in each clade. Notably, all seven clades were present and detectable at the very beginning of the pandemic albeit with variable abundancy. The Alpha clade was the first to strongly emerge and spread worldwide, while Beta/Mu and Lambda/Gamma remained at low levels of diffusion. Regarding their respective CUBs, all clades exhibited very similar patterns during the first 3 months of 2020. From April to September 2020, the RSCU pattern of the Alpha clade began to fluctuate markedly before reaching a long period of stability from October 2020 to September 2021, coinciding with the peak of Alpha clade expansion worldwide. Surprisingly, the CUB pattern during this high transmission period was reversed compared to the initial pattern. The pattern of a stabilized, reversed CUB was observed not only in the Alpha clade but also in the Delta and Omicron clades when they both reached their highest levels of transmission. 

The clear correlation between CUB inversion and the peak of diffusion can be interpreted in two opposite ways: (a) the high level of expansion has selected sub-variants with a stable CUB, or (b) a highly adapted sub-variant with optimized CUB preceded the peak, driving the exponential expansion. Although both scenarios might be involved, the persistence of an inverted CUB beyond peak diffusion suggests natural selection of an optimized CUB likely played a role (favoring option b).

Comparing the RSCUs of both genes at peak expansion for each clade reveals that Spike exhibits a much stronger preference for codons ending by A/U, especially those recognized by tRNAs modified at the U_34_ position (e.g., Arg_AGA_, Lys_AAA_, Glu_GAA_, and Gln_CAA_) (Figure 7B). This necessitates analyzing RSCU patterns for each gene individually at expansion peaks. Interestingly, each clade shows a different CUB optimization pattern during peak diffusion, indicating that multiple codon patterns can support an optimal viral translation (Figure 7C,D). For instance, Omicron has evolved a codon bias diametrically opposed to the CUB of the other clades, while the Alpha and Delta Spike sequences share similar CUB patterns (Figure 7C). Conversely, the Orf1ab gene in the Alpha and Gamma/Lambda clades show the closest patterns (Figure 7D). Furthermore, within a given clade, the Orf1ab and Spike genes have developed distinct RSCU patterns. These varying preferences for A/U ending codons or codons requiring U_34_-modified tRNAs suggest that efficient translation might depend on subtle but specific factors like translation speed and protein-folding parameters.

### 2.4. Hypothesis: Virus-Driven Manipulation of the tRNA Pool and tRNA Modifications Forces Translation of SARS-CoV-2 Genes

Bioinformatic approaches like the tAI and the CAI measures reasonably predict gene expression but can be improved. More precise estimations of amino acid-loaded tRNA (“ready-to-translate” tRNAs) availability would be more realistic than using the concentration of tRNA molecules or its estimate from the tRNA gene copy number. The real availability of functional tRNAs, influenced by tissue-specific pools, is required for high translation efficiency. Another important mechanism for efficient protein synthesis is the channeling effect, which involves the direct transfer of aminoacyl-tRNA/tRNA between aminoacyl-tRNA synthetases (ARS), elongation factors, and ribosomes. This process is crucial because it regenerates and concentrates frequently used tRNAs near specific translation sites [140,141]. Additionally, the global CUB measures (CAI, tAI, RSCU) do not consider the order of favorable and unfavorable codons along the transcript, which can create fast or slow translation segments [41,106]. Environment-dependent dynamic variations in the tRNA pool and tRNA demand should also be integrated into future models of translation efficiency. Using tRNA composition from cancer cells can distort the prediction of virus/host-tissue translational compatibility due to altered tRNA levels [142] and tRNA modifying enzymes in pathological conditions [143]. U_34_ modifications, in particular, rapidly respond to metabolic changes [144] such as methionine metabolism, carbon balance, or phosphate homeostasis [145,146,147].

Translational reprogramming emerges as a key element in cell adaptation to changing environments [148] and may aid virus adaptation to hosts. We have here highlighted striking discrepancies between SARS-CoV-2’s preference for U_34_-sensitive codons and the availability of cognate U_34_-modified tRNAs (i.e., mcm^5^s^2^) in target cells. Since SARS-CoV-2 has not fully adjusted its codon bias to match human target cells, its recent rapid expansion may be due to its ability to manipulate U_34_ modifications, optimizing translation and facilitating its infection cycle. By reducing the need for precise codon usage adaptation, this ability could allow the virus to infect a broader range of hosts [149].

### 2.5. Evidence Supporting the Ability of SARS-CoV-2 to Exploit the tRNA Epitranscriptome to Favor Viral Translation

#### 2.5.1. Potential Manipulation of tRNAs by SARS-CoV-2

Direct evidence supporting this hypothesis for SARS-CoV-2 is currently limited in the literature, except for a recent study by Tao Pan’s group [69], which reported the presence of cellular tRNA in SARS-CoV-2 particles. This study suggested that each virion contains at least four different tRNA molecules. Notably, among the eight tRNAs preferentially enriched in SARS-CoV-2 particles, 75% require U_34_ modification for efficient decoding, including tRNA^Lys^_UUU_ and tRNA^Glu^_UUC_, both bearing the mcm^5^s^2^U_34_ modification. This observation suggests preferential packaging of critical tRNAs complementing skewed SARS-CoV-2 codons, reinforcing the assumption that U_34_-sensitive codons in viral genes require adaptation of the host tRNA pool for efficient viral translation. Other enzymes involved in tRNA modifications might also be involved in this virus-mediated translational control, such as the tRNA methyltransferase TRMT1 that generates the m^2^_2_G_26_ mark. Indeed, recently, we [70] and others [71,72] have shown that TRMT1 was specifically proteolyzed by the SARS-CoV-2 Nsp5 main protease leading to a decrease in the m^2^_2_G_26_ modification on tRNAs in infected cells, negatively impacting viral replication. This manipulation, suggests the role of m^2^_2_G_26_ tRNA modification patterns in cellular pathogenesis and biology of SARS-CoV-2 infection.

This situation is not unique to SARS-CoV-2. Recent studies have shown that during the Alphavirus CHIKV infection, deregulated expression of KIAA1456, an ALKBH8 homolog able to generate the mcm^5^U_34_ modification of tRNAs, consequently reprogramed codon optimality and favored viral RNA translation [30]. This mechanism was also shown to occur during the Flavivirus DENV infection, which, like CHIKV, exhibits a strong enrichment in U_34_-sensitive codons [30,73]. In parallel, a recent preprint [150] proposed that DENV is also able to exploit host tRNA by a different mechanism involving the ALKBH1 RNA modifier and the virally encoded NS5 protein (with dual RNA methyltransferase (MTase) and RNA-dependent RNA polymerase (RdRp) enzyme activities). The cellular and viral enzymes can both mediate f^5^Cm-modification of the cytoplasmic tRNA-Leu(CAA) at the wobble position C_34_ and collaboratively promote pro-viral translational remodeling during DENV infection.

#### 2.5.2. Experimental Data Revealing SARS-CoV-2-Induced tRNA Epitranscriptome Modulations

To assess the validity of our hypothesis, we investigated the dynamic changes in tRNA modifications within SARS-CoV-2-infected cells. We first explored by LC-MS/MS the behavior of epitranscriptomic marks in the tRNA subpopulation extracted from SARS-CoV-2-infected VeroE6 cells (Figure 8A). Using this approach, we tracked 21 different tRNA post-transcriptional modifications and discovered that modifications at position 34, including ncm^5^U, mcm^5^U, and mcm^5^s^2^U, were noticeably increased. We extended our analysis to human Caco2 cells, a more suitable cellular model for exploring SARS-CoV-2 infections given the known COVID-19 gastrointestinal manifestations. Our focus on U_34_ modifications once again highlighted the early and rapidly changing nature of the three marks we examined (Figure 8B). By simultaneously tracking tRNA levels, we also found that two of the four U_34_-modified tRNAs (tGlu^UUG^ and tGln^UUC^) were upregulated at later time points (Figure 8C). Overall, these preliminary results indicate the possible role of tRNA modifications in SARS-CoV-2 infection and reinforce the idea that SARS-CoV-2 can manipulate the host’s tRNA transcriptome. However, it remains unclear whether these changes can genuinely benefit viral infection. 

#### 2.5.3. SARS-CoV-2 Infection Is Impaired When the tRNA U34 Modification Pathway Is Disrupted 

In mammals, enzymes responsible for the chemical U_34_ modification include the Elongator complex (Elp1–6), Alkylation repair homolog 8 (Alkbh8), and thiouridylases (Ctu1/Ctu2) [28]. If the induction of ncm^5^U, mcm^5^U, and mcm^5^s^2^U marks indeed benefits viral translation, virus infection should be closely tied to the accurate activity of the Elongator complex. Using primary fibroblasts from patients with Familial Dysautonomia (FD) (Figure 9A), which lack Elp1 expression (=IKBKAP^−/−^) and consequently have reduced levels of all three U_34_ tRNA modifications (Figure 9B) [28,151], we observed significantly lower levels of SARS-CoV-2 infection in FD cells compared to control fibroblasts (*wt*) from healthy individuals (Figure 9C). These preliminary data emphasize the critical role played by U_34_ modifications of host tRNAs in the SARS-CoV-2 lifecycle and provide the first evidence of SARS-CoV-2’s ability to rewire the tRNA epitranscriptome to facilitate translation of its heavily codon-biased genome.

## 3. Concluding Remarks

### 3.1. Is Altering tRNA Epitranscriptome a Common Viral Strategy?

Viruses are highly dependent on the host cell’s translation machinery, including host tRNAs, for the efficient translation of their genetic material [41,152,153], and this process is heavily influenced by chemical modifications of tRNAs, catalyzed by various tRNA-modifying enzymes, particularly within the tRNA anticodon loop region [27,154]. We here showcased SARS-CoV-2’s ability to directly target this crucial step by upregulating U_34_ tRNA modifications to facilitate the translation of its genome, which is enriched in U_34_-sensitive codons. In addition to Chikungunya and Dengue, for which the ability to interfere with the host tRNA epitranscriptome has been suggested [30,73,150,155], Zika can have also evolved the same tactic to overcome its high degree of preference for U_34_-sensitive codons [58]. In addition to tRNA modification, translation efficiency favoring viral translation can also be manipulated by altering tRNA levels [156], as we briefly illustrated in the case of SARS-CoV-2. Retroviruses such as HIV [43] or DNA viruses such as SV40 [157], EBV [158], Adenovirus [159], and HSV-1 [160] are able to manipulate tRNA levels by stimulating Pol III transcription of tRNA genes.

### 3.2. Future Priority Investigation Areas

New approaches to explore the dynamics of the tRNA epitranscriptome during viral infections are imperative. The direct quantification of tRNA pools with new experimental approaches like the recently developed mim-tRNAseq, which simultaneously measures tRNA abundance and modifications [161,162] could provide insight alongside Ribo-seq analysis of viral and cellular translatomes. Expansion of the toolbox for quantitative recording and understanding the chemical biology of the tRNA epitranscriptome is clearly needed. This will include emerging technologies for mass spectrometry-based [163,164] and nanopore-based tRNA modification mapping [165,166] and analysis of ribosome-bound tRNAs [167]. In this respect, MLC-seq [168] (mass spectrometry ladder complementation sequencing), a recent groundbreaking mass spectrometry approach, offers a potential solution to these challenges by providing quantitative, and site-specific mapping of RNA modifications, revealing the truly complete informational content of tRNA. The recent development of DORQ-seq [169], a combination of cDNA hybridization and deep sequencing, will also deliver a detailed tRNA composition matrix from femtomolar amounts of total tRNA. Advances in nanopore sequencing are anticipated with optimized basecalling models that could allow enhanced detection of RNA modifications and mapping [170].

The essential role of tRNA modifications in translation regulation is emerging as a new crucial host process hijacked by RNA viruses to adapt host translation to their sub-optimal codon bias. It is therefore decisive to investigate the underlying mechanisms involved, particularly those targeting U_34_ tRNA modifications. By genetically inactivating U_34_ enzymes using CRISPR/Cas9 or RNA interference, we can assess their impact on SARS-CoV-2 translation and replication. Additionally, exploring how individual viral proteins influence the abundance and activity of U_34_ enzymes can provide valuable insights into the mechanisms of this new level of virus-host interaction.

Technological and scientific advances in RNA modification highlighted their role in viral RNA structure, localization, splicing, stability, and translation [171,172]. Understanding viral- or host-induced alterations of RNA marks is essential for understanding gene regulation, identifying essential marks for the virus cycle, and designing appropriate drugs. Nucleoside-derived inhibitors targeting SARS-CoV-2 nsp14 (N7-guanine)-methyltransferase have recently shown promise in crippling the stability of viral RNA [173]. Exploring virus–tRNA epitranscriptome interactions could open promising new avenues for therapeutic intervention.

## 4. Material and Methods

### 4.1. Bio-Informatics—Codon Analysis

Codon adaptation index (CAI) and Relative Synonymous Codon Usage (RSCU) calculations were carried out using CAIcal web-available tools (http://genomes.urv.es/CAIcal, accessed on 29 January 2020) [99]. Nc data were obtained using CAIcal and plotted with Rstudio-ggplot2. Codon frequencies were calculated using the Codon Utilization Tool (CUT) website (http://pare.sunycnse.com/cut/index.jsp, accessed on 8 September 2021) [174]. tRNA adaptation index (tAI) was calculated using stAIcalc software version 1.0 [107] (http://www.cs.tau.ac.il/~tamirtul/stAIcalc/stAIcalc.html, accessed on 11 July 2023). Cluster analysis was carried out using Genesis 1.8.1 [175] and Cluster 3.0 [176] and visualizations were made with Java Treeview (Version: 2.11.4.0; https://www.jalview.org/, accessed on 5 June 2021) [177]. Protein abundance levels were derived from PaxDb database version 4.1 (https://pax-db.org/, accessed on 27 August 2023) [178]. 

To verify the accuracy of the interpretation of *tAI*, we compared the consistency in tRNA expression across different human primary cell types by visually examining the relationship between tRNA gene copy number (GCN) and experimentally measured tRNA levels in various human primary cell types. We used a previously published dataset from the Nedialkova group [114] including additional information concerning modifications at position 34 occurring on some specific tRNAs.

### 4.2. SARS-CoV-2 Sequences

Accession numbers of the main SARS-CoV-2 sequences used in this report are listed in Appendix A and were downloaded from the NCBI database with the exception of SARS-CoV-2 FRA, which was downloaded from the European Virus Archive (http://www.european-virus-archive.com, accessed on 8 June 2020) and corresponded to the exact 2020 isolate from Paris-Ile-de-France that we used in our infection experiments. Analysis of clade-related RSCU evolution of SARS-CoV-2 during the pandemic was performed using sequences recovered from the dataset generated in the Fumagalli group’s paper [68].

### 4.3. Experimental Data

#### 4.3.1. Cells and Viruses

Patient primary fibroblasts were from the Coriell Institute: GMO1652 derived from non-FD control (Skin fibroblast (arm) from 11-year-old Caucasian female); GMO4959 derived from FD patient (Skin fibroblast (arm) from 10-year-old Caucasian female). The SARS-CoV-2 was a French Ile de France isolate (www.european-virus-archive.com/virus/sars-cov-2-isolate-betacovfranceidf03722020, accessed on 8 June 2020). Viral stocks were generated by amplification on VeroE6 cells (epithelial kidney of an African green monkey, ATCC CRL-1586). The supernatant was collected, filtered through a 0.45 µm membrane, and titered using a TCID50 assay. Caco2 cells (epithelial colon adenocarcinoma, ATCC HTB-37) were used for tRNA modification quantification upon SARS-CoV-2 infection.

#### 4.3.2. Quantification of tRNA Modifications by Mass Spectrometry (LC-MS/MS)

RNA preparations enriched in tRNAs were obtained using mirVana™ miRNA Isolation Kit (Thermo Fisher Scientific, Waltham, MA, USA). RNA samples were then digested by Nuclease P1 and treated by Alkaline phosphatase. Samples were then injected into an LC-MS/MS spectrometer. The nucleosides were separated by reverse phase ultra-performance liquid chromatography on a C18 column with online mass spectrometry detection using an Agilent 6490 triple-quadrupole LC mass spectrometer in multiple reactions monitoring (MRM) positive electrospray ionization (ESI) mode. Quantification was performed by comparison with the standard curve obtained from pure nucleoside standards running with the same batch of samples.

#### 4.3.3. Assays for Viral Replication

For infections, the cells were previously transduced with a Lentiviral vector expressing ACE2 using the lentiviral construct RRL.sin.cPPT.SFFV/Ace2.WPRE (MT136) kindly provided by Caroline Goujon (Addgene plasmid # 145842) [172]. Seventy-two hours after transduction, accurate ACE2 expression was controlled on Western blot probed with anti-ACE2 antibody (Human ACE-2 Antibody, AF933, R&D systems). ACE2-positive cells (70–80% confluence) were then infected with SARS-CoV-2 diluted to achieve the desired MOI. After 24 h culture, the cells were lysed with the Luna cell-ready lysis module (New England Biolabs, Ipswich, MA, USA). The amplification reaction was run on a LightcyclerR 480 thermocycler (Roche Diagnostics, Indianapolis, IN, USA) using the Luna Universal One-Step RT-qPCR kit (New England Biolabs), and SARS_For: 5′-ACAGGTACGTTAATAGTTAATAGCGT; SARS_Rev: 5′-ATATTGCAGCAGTACGCACACA; GAPDH_For: 5′-GCTCACCGGCATGGCCTTTCGCGT and GAPDH_Rev: 5′-TGGAGGAGTGGGTGTCGCTGTTGA primers. Each qPCR was performed in triplicate, and the means and standard deviations were calculated. Relative quantification of data obtained from RT-qPCR was used to determine changes in SARS-CoV-2 gene expression across multiple samples after normalization to the internal reference GAPDH gene. Individual tRNA quantification was performed by RTqPCR using tRNA-gene-specific primers and miR103a primers for internal normalization purposes (see Appendix B).

## Data Availability

All data generated or analyzed during this study are included in the manuscript or cited in reference.

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
