# Peer review of "SARS-CoV-2 Displays a Suboptimal Codon Usage Bias for Efficient Translation in Human Cells Diverted by Hijacking the tRNA Epitranscriptome"

_ijms, 2024, doi:10.3390/ijms252111614_

Round 1

Reviewer 1 Report

Comments and Suggestions for Authors

The study is well-written. The study focuses on an extensive analysis of the codon bias in SARS-CoV-2 and how it influences translation efficiency in human cells. The authors investigated the role of codon usage, tRNA modifications, and translational efficiency using both bioinformatic and experimental approaches.

Here are some minor corrections :

- Results and Discussion Section:

*Authors wrote, "Several reports based on state-of-the-art bioinformatics suggest that SARS-CoV-2 originated from an ancestral coronavirus in bats, specifically related to the bat RaTG13 coronavirus, that likely passed through intermediate hosts such as pangolin (Manis javanica) before crossing species barriers again to infect humans".   It would be clearer and more precise if the authors specified the exact bioinformatics methods or tools employed in these studies instead of writing  'state-of-the-art bioinformatics'.

- Materials and Methods:

* Please write the version of Java Treeview.

Author Response

We are appreciative of reviewer 1's constructive evaluation of our manuscript and the valuable corrections he/she recommended.

Comment 1

Results and Discussion Section:

*Authors wrote, "Several reports based on state-of-the-art bioinformatics suggest that SARS-CoV-2 originated from an ancestral coronavirus in bats, specifically related to the bat RaTG13 coronavirus, that likely passed through intermediate hosts such as pangolin (Manis javanica) before crossing species barriers again to infect humans".   It would be clearer and more precise if the authors specified the exact bioinformatics methods or tools employed in these studies instead of writing  'state-of-the-art bioinformatics'

Response 1:

  • Stat-of-the art bioinformatics was replaced by: sequence, phylogenetic and recombination analyses.

Comment 2:

Please write the version of Java Treeview

Response 2:

The version of Java Jalview Treeview we used and the weblink was added:

"Java Treeview (Version: 2.11.4.0; https://www.jalview.org/)"

Reviewer 2 Report

Comments and Suggestions for Authors

The topic of this manuscript is interesting and the hypothesis proposed by the authors is appealing. However, I am not sure if the way the paper is presented is appropriate. It reads like a review, but at the same time, it is an experimental paper.  If authors want to show data a support a hypothesis, even if this is theoretical, they should write a manuscript like a regular research paper. Otherwise, it looks confusing. 

Comments on the Quality of English Language

English is fine with minor edits required... eg. space should be deleted after Arg in line 3 of Abstract, etc.

Author Response

We acknowledge the reviewer's concern that the article's format, which straddles the boundaries of review and experimental reporting, could be perplexing to the reader. The manuscript's structure adheres to the IJMS 'Hypothesis' guidelines we've agreed upon with the special issue editors, Letrari Liu and Rosemary Kiernan, given the original in silico and experimental in vitro data it contains. 

Comment 1 ( I am not sure if the way the paper is presented is appropriate)

Response 1:

  • To improve the article's readability, we’ve added the following concise summary of its unfolding at the end of the Introduction section 1:

“We here present a re-evaluation of SARS-CoV-2 coding sequences, specifically examining the congruence between viral codon preferences and the composition of the host's tRNA pool. Our analysis, using a specialized codon analysis toolkit, found that SARS-CoV-2's codon demand doesn't perfectly match the human host's tRNA supply. Given that the codon bias was predominantly attributable to an enrichment in A-ending codons, which necessitate RNA modifications of the cognate tRNAs at position U34 for efficient decoding, we hypothesized that the virus might manipulate the host's tRNA epitranscriptome to circumvent this translational incompatibility. This hypothesis was subjected to experimental validation by investigating the impact of SARS-CoV-2 infection on tRNA modifications, revealing a substantial enhancement of U34 tRNA modifications upon infection. The additional observation that human cells unable to properly modify U34 tRNAs are no longer able to fully support the replication and the sustained spread of SARS-CoV-2, confirmed that SARS-CoV-2 infection efficiency depends on the accurate activity of the U34-tRNA modification pathway.”

From IJMS Instruction to Authors

Hypothesis

Hypothesis articles introduce a new hypothesis or theory, or a novel interpretation of that theory. They should provide: (1) a novel interpretation of recent data or findings in a specific area of investigation; (2) an accurate presentation of previously posed hypotheses or theories; (3) the hypothesis presented which should be testable in the framework of current knowledge; and (4) the possible inclusion of original data as well as personal insights and opinions. If new data are presented, the structure should follow that of an article. If no new data are included, the structure can be more flexible, but should still include an Abstract, Keywords, Introduction, Relevant sections, and Concluding Remarks.

Comment 2 (English is fine with minor edits required... eg. space should be deleted after Arg in line 3 of Abstract, etc.)

Response 2:

  • The space after Arg in line 3 of Abstract was deleted: "a strong preference for LysAAA, GlnCAA, GluGAA, and ArgAGA infrequently used in human..."

Round 2

Reviewer 2 Report

Comments and Suggestions for Authors

None

Comments on the Quality of English Language

None

Author Response

We thank Reviewer 2 for his/her observations. We check the overall text for English language editing.

Best regards,

Patrick Eldin.